# An Integrated Kano Model, Fuzzy Analytical Hierarchy Process, and Decision Matrix for Sustainable Supplier Selection in Palm Oil Industries Indonesia, a Case Study

Tsai-Chi Kuo [1,2], Muniroh Muniroh [1,*] and Kristin Halisa Fau [3]

[1] Department of Industrial Management, National Taiwan University of Science and Technology, Taipei 106335, Taiwan; tckuo@mail.ntust.edu.tw

[2] Artificial Intelligence for Operations Management Research Center, National Taiwan University of Science and Technology, Taipei 106335, Taiwan

[3] Department of Industrial and Systems Engineering, Chung Yuan Christian University, Taoyuan 32023, Taiwan; kristinhalisafau@students.itb.ac.id

\* Correspondence: m10501836@gapps.ntust.edu.tw.com; Tel.: +886-2-2730-3277

**Abstract:** Industries have to integrate environmental, social, and economic aspects into their supply chain management to achieve sustainability. Hence, the industry needs to take appropriate actions in choosing the right suppliers. The aim of this study is to develop a framework for selecting sustainable suppliers by integrating quality management tools using the Kano model, Fuzzy Analytical Hierarchy Process, and Decision Matrix Method. To identify the critical sustainability criteria, the Kano model by the clustering the criteria for sustainable selection supplier was used. We then used the Fuzzy Analytical Hierarchy Process to determine the weight of each criterion and applied the Decision Matrix Method to select the most sustainable supplier. Afterward, the appropriate proposed framework was implemented in one of the palm oil industries in Indonesia to validate that the framework is applicable and useful. The study shows that the environmental dimension is the most sustainable supplier criteria followed by economy and social dimension. Quality, pollution control, and information disclosures were found to be important sub-dimensions for sustainable supplier selection.

**Keywords:** sustainability; sustainable supplier selection; the Kano model; Fuzzy Analytical Hierarchy Process; Decision Matrix Method





## 1. Introduction

The increasing awareness about sustainability and the existence of government regulation has forced the industry to include sustainability in their manufacturing process. Hence, industries are expected to transform their business process into sustainable and to afford a positive impact on the environment while also paying attention to social and economic issues [1]. The operation performance of the supply chain is linked to the whole business process. Thus, the best way to make a significant transformation is by managing the economic, environmental, and social impacts of supply chains [2]. The integration of sustainability attributes towards supply chain management practices is referred to as sustainable supply chain management. Sustainable supply chain management is also known as a system and a type of work that has a social responsibility that contributes to economic development and maintains good business ethics [3]. Liu et al. [4] revealed that sustainable supply chain management is a traditional supply chain that is transformed due to the widespread attention on environmental and green issues.

Implementing a sustainable supply chain can create significant implications to supply chain members, especially for suppliers. The role of suppliers will directly influence industry efforts to achieve sustainability [5]. In addition, supplier performance also relates to industry performance. While suppliers have to implement a policy regarding sustainability

into their supply process, the industry needs to take appropriate action in choosing the right supplier to be a work partner and aligned with the company [6]. The anticipated suppliers are those who implement sustainability in their process and support the industry to achieve sustainability, which is called a sustainable supplier.

Through considering sustainability as a popular issue these days and the importance of selecting a supplier that can support industry to achieve sustainability to their business process, this study concerns performing an in-depth study of supplier selection. In previous research, supplier selection who considered economic, environmental, and social aspects, are scarce. Rashidi et al., [7], who conducted a meta literature review about sustainable supplier selection, revealed that from the year 2010 to 2014, only about 3 to 4 papers related were published each year. The trend started to increase between 2015 to 2019, where around 8–19 articles were published. Mostly, it only considers economics along with ecological aspects. Previous studies usually identify and select criteria based on literature review and interview the expert. Other than that, they only used one method or integrated the MCDM method with fuzzy techniques. However, this research attempts to integrate quality management tools, MCDM methods, and fuzzy techniques to develop a framework for selecting a sustainable supplier. Those tools are intended to build a robust methodology, so the developed framework is valid and applicable.

In this study, the Kano Model is used to identify the critical sustainability criteria by clustering processes, then the Fuzzy Analytical Hierarchy Process is applied to determine the weight for each criterion, and the Decision Matrix Method is used to select the most sustainable supplier among several alternatives. Afterward, the proposed framework is implemented in the real world to validate that the framework is appropriate and efficient. The palm oil industry in Indonesia is selected as a case study due to the finding that the industry has implemented sustainability in its process [8]. Indonesia itself is the biggest producer and exporter of palm oil worldwide. More than seventy percent of total production is exported to international markets, with the central export destination countries being China, India, Europe, Pakistan, and Malaysia [9]. In 2011, Indonesia's government established a policy called Indonesian Sustainable Palm Oil (ISPO), whose goal is to enhance the global competitiveness of Indonesian palm oil and also to increase concern for environmental issues [8]. ISPO is officially issued by the Directorate General of Plantations, and this standard is already followed and mandatory because it is based on Indonesian government laws and regulations [8]. This standard must be implemented for plantation business actors in Indonesia and the target of its implementation begins in 2012 [10].

The environmental impact of the oil palm industry is a serious threat in the midst of the Mahakam Wetlands, Kalimantan [11]. Pollution from the palm oil industry includes agricultural chemicals related to biofuels (fertilizers, pesticides and rodenticides) which have a harmful impact on terrestrial and aquatic ecosystems [12]. The decreasing oxygen levels and increasing nitrate loading (eutrophication) are correlated with the burgeoning palm oil industry in the region and its effect on the Lakes nearby in Kalimantan [13]. Additionally, the wildfire smoke pollution from the expansion of the palm oil industries adversely affects human health and productivity in Southeast Asia [14]. In Katapang, Indonesia, fire was the cause of 90% of deforestation between 1989 and 2008 [14], and 20% of wildfires across Indonesia can be attributed directly to oil palm plantation practices [15]. The creation and enforcement of environmental standards is critical to the success of managing the sustainable development of the palm oil industry. Therefore, environmental impacts of new development must be carefully considered. Other than that, based on the literature review, it was rarely found a study using the palm oil industry as a case study. Therefore, it can be stated that the palm oil industry is concerned about sustainability matters and has implemented it into their business process to achieve sustainability goals. The outlines of the paper are as follows: Section 1 Introduction. Section 2 Literature review. Section 3 proposed an integrated methodology. Section 4 contains the results of the study, and the last Section 5 discussion of the study.

## 2. Literature Review

All aspects of the members in the supply chain should be sustainable, including partners of the company. Suppliers that are environmentally friendly, socially, and economically concerned are very crucial in the implementation of sustainability, which can lead to improved performance in supply chain management [16–18]. Previously, supplier selection only considered economic aspects [19,20]. Nevertheless, due to globalization, competitive market situation, and changing market demand, the consideration for supplier selection was shifting [21]. Nowadays, supplier selection also considers environmental and social aspects that developed into sustainable supplier selection. The first supplier selection that incorporated environmental criteria had been conducted in 1977 by Noci. Further, many researchers considered environmental and economic criteria in supplier selection research study from Bhutta [22]; Gurel et al. [23]; Igarashi et al. [24] and considered three dimensions of sustainability in supplier selection as in a study by Bai et al. [25]; Tavana et al. [26].

Sustainable supplier selection is a process to identify and evaluate suitable suppliers and their upstream supply chain regarding the three dimensions of sustainability. There are two essential issues in sustainable supplier selection. First is the criteria and sub-criteria for performance evaluation, and second is the approach or method for selecting the best supplier [27]. Mahmood et al. [28] investigated 143 peer-reviewed publications on sustainable supplier selection from 1997 to 2014. They obtained the top 10 criteria for three-dimension on sustainability as follows: (1) Economic criteria: Quality, price, flexibility, cost, logistics costs, lead time, relationship, technical capability, reverse logistics, rejection ratio (ppm).; (2) Environmental criteria: Environmental management system, recycling, controlling ecological impacts, resource consumption, eco-design, energy consumption, wastewater, reuse, air emissions, environmental code of conduct' (3) Social criteria: Involvement of stakeholders, stakeholder relations, the rights of stakeholders, staff training, health and safety, safety practices, social code of conduct, social management commitment, donations for sustainable projects, and the annual number of accidents [28].

In literature, most authors perform a case study in different industries by using their proposed methodology for supplier selection, for instance: dairy company, automobile company, iron & steel company, electronic company, importing company, wood industry, home appliances manufacturer, etc. Sustainable supplier selection involves more than one criteria, and these criteria often conflict with each other. Therefore, it becomes a complex and multicriteria decision-making (MCDM) problem. The MCDM method is effective in dealing with decision-making processes of complex problems yet additionally enables DM's to consider and adjust the trade-offs among a broad scope of criteria that can influence a choice [29]. Tavana et al. [26] applied an integrated Analytic Network Process (ANP) and Quality Function Development (QFD) for sustainable supplier selection. Guner [30] integrated the Fuzzy Decision Making Trial and Evaluation Laboratory (DEMANTEL) for calculating the weights of sustainable criteria and Taguchi loss functions for evaluating and ranking sustainable suppliers in supplier selection. Tamirat et al. [31] employed a novel approach using the process yield index and proposed a Bonferonni correction method to measure product quality in sustainability supplier selection.

There are many sustainability criteria used in selecting suppliers, creating a challenge for decision-makers in evaluating the performance of suppliers, and so quality attributes have been assumed to denote the importance weight of the criteria for supplier selection [32]. With these considerations, the authors proposed a Kano model to classify and categorize the sustainability criteria based on needs and priorities. The Kano model is intended to categorize the attributes of a product or services based on the ability to fulfill and satisfy customer needs [33]. There are three steps in the Kano process: Kano questionnaire, Kano evaluation table, and Kano category result [34]. Lee and Huang [35] confirmed that the Kano model analysis is an effective mechanism for analyzing customer needs. The Kano model has a functional and dysfunctional question for each attribute. It identifies the needs of decision-makers regarding the fulfillment of product attributes and categorizes those attributes into different clusters [36]. The number of researchers in the

Kano model has been rising, the research content has been deepening, and the influence of the Kano model has been expanding. Arabzad et al. [37] introduced an integrated Kano-DEA model for distribution evaluation problems in the supply chain. Mazaher et al. [32] applied a novel oncoming for supplier selection using the Kano model and fuzzy MCDM. The proposed method has three phases and is implemented in an agricultural machinery manufacturing factory in Iran for validation. Kilaparthi [38], in her study, constructed a hybrid methodology which comprises Data Envelopment Analysis (DEA), Kano Model Analysis, and VIKOR for selecting and evaluating the best supplier for a firm. Jain and Singh [36] used an integrated method by Fuzzy Kano and Fuzzy Inference System for selecting sustainable suppliers.

Analytic Hierarchy Process (AHP) is stated as a successful theory because of the consistency of its assumptions with available experimental data. It establishes the predictions able to be tested based on experiments, and it explains the behavior [39]. Based on that, AHP is the suitable tool for solving the supplier selection question, which involves a lot of intangible factors, but still needs a logical and rational control of decisions. Fuzzy AHP is developed from classical AHP, which considers the fuzziness of the decision-makers. Kahraman et al. [40] revealed that Fuzzy AHP methods are structural approaches for selection and evaluation problems by using the concepts of fuzzy set theory and hierarchical structure analysis. Kannan et al. [41] studied a hybrid model to select green suppliers by using fuzzy TOPSIS. The objective of their model is to concurrently maximize the total value of purchasing and to also minimize the total cost of purchasing. Shaw et al. [42] presented an integrated approach using fuzzy AHP and fuzzy multi-objective linear programming for supplier selection and addressing the carbon emission issue. The proposed method is efficient for handling a realistic situation with the involvement of vague information. Awasthi et al. [43] applied fuzzy AHP for generating criteria weights for multi-tier sustainable global supplier selection. Mohammed et al. [44] used Fuzzy AHP to establish important weights of the criteria for sustainable two-stage supplier selection and order allocation problems. The benefit of Fuzzy AHP is the ability to manage uncertainty and ensure consistent ranking from the judgment by using pairwise comparisons [45,46]. Ishizaka [47] revealed Fuzzy AHP method is effective in identifying an appropriate supplier and evaluating its performance.

A decision matrix is best used when assessing an alternative from a rational perspective and has enough comparable variables to make a weighted analysis. Multicriteria decision making (MCDM) is dealing with decisions, demanding the best alternative choices from several possible candidates in a decision, concerned with certain criteria that may be concrete or vague [48]. Sustainable supplier selection requires multicriteria decision-making (MCDM); thus, the decision-maker needs to consider and evaluate qualitative and quantitative factors. Wang et al. [49], in their study, applied the multicriteria group decision-making (MCGDM) model for supplier selection in a rice supply chain. Gonçalo and Morais [50] applied a multicriteria group decision method for oil supplier selection in Brazilian oil companies. By using the Preference Ranking Organization Method for Enrichment Evaluation (PROMETHEE II), they acquire decision-makers individual evaluations and the voting procedure by quartiles so as to convert the individual positions into a position for the group.

Based on the literature review, a supplier selection study that considers three pillars of sustainability is scarce. Most previous research only focuses on economic criteria. As time goes by, some research begins to concern environmental aspects in supplier selection along with economic criteria. The previous study identifies and selects criteria based on a literature review or interviews with an expert. The author found only three papers which implemented criteria clustering: Mazaher et al. [32]; Jain and Singh [36]. The purpose of performing criteria clustering is to figure out the must-have criteria for a sustainable supplier based on industry satisfaction and for better evaluation when there are many sustainability criteria used in supplier selection. According to that, the author proposed a

Kano model for criteria clustering to capture the palm oil industry needs and fulfill their satisfaction in selecting a sustainable supplier.

Indonesia is the biggest palm oil producer in the world. In 2011, Indonesia Government made it mandatory for all palm oil industries to have a certification scheme known as Indonesia's Sustainable Palm Oil (ISPO) [8]. Those certifications became a trigger to all palm oil industries to be more aware and concerned about sustainability in their business process. Nowadays, most palm oil industries not only increase their production to achieve profit but also take into consideration to respect the social interest and protect the environment. According to that description, this research selected the palm oil industry in Indonesia as a case study. One of the best ways to support sustainability in the palm oil industry business process is by selecting sustainable suppliers. In addition, the research about sustainable supplier selection using empirical study in the palm oil industry was rarely found in previous research.

This study proposed a novel integrated method for sustainable supplier selection by integrating quality management tools, fuzzy techniques, and the MCDM method. Fuzzy AHP was used to determine the weights of each criterion due to the ability of that method to handle uncertainty, vagueness, and ambiguities of decision-makers in the decision-making process. Following that, the author conducted a case study to show the proposed model is valid and applicable. A decision matrix is used to determine the most sustainable suppliers. The decision matrix was chosen because it was simple, easy to use, and useful for decision-making with several alternatives. Therefore, it can be stated that the proposed integrated method is able to capture industry needs, perform well with uncertainty, and be uncomplicated in the application.

## 3. Materials and Methods

### 3.1. Data Collection

The author designed and distributed the Kano questionnaire to several respondents to collect the data for clustering the sustainability criteria. The qualification for the respondents is someone who works in the palm oil industry; his work relates to suppliers and has a piece of knowledge about sustainability. That qualification is to make sure the result obtained in the clustering process is specifically intended for the palm oil industry and also to find priority criteria based on the industry's need for selecting a sustainable supplier. During the survey, thirty-five respondents filled the questionnaire. However, there are two questionnaires excluded due to inappropriate data. Therefore, there were thirty-three valid filled questionnaires used as data and to be processed through the Kano model (Appendix A). Furthermore, the thirty-three respondents were employees in the palm oil industries around Indonesia and worked in quality control, sustainability, purchasing, production, marketing, quality assurance, accounting stock, finance, industry relation, and HRD department. The result of the clustering process through the Kano model is completely based on the respondent's preferences and assessments. For that reason, the obtained results are expected to be suitable with the palm oil industry's need for a sustainable supplier. Thus, it can be stated that the criteria in this study are specific and different compared to other studies due to different industry needs.

As for pairwise comparisons, the author also conducted a second phase online questionnaire to collect data from the respondents. Later, this data were used to obtain the weight of each "must be" sustainability criteria. The respondent must be an expert who has worked and experienced in one of these field areas: purchasing, environmental, production, logistic, marketing, finance, and quality assurance in the palm oil industry. Rating for decision matrix, the author distributed a decision matrix online form to respondents to obtain the most sustainable supplier from several alternatives. In this step, three suppliers are evaluated.

### 3.2. Research Development

In this research, there are four analysis steps used in this study as follows: identify the criteria for sustainable supplier selection, cluster the sustainability criteria using the Kano model, determine the weight of each "must be" criteria using Fuzzy AHP, and select sustainable supplier using Decision Matrix. Figure 1. presents the research design of this study. The criteria for sustainable supplier selection are constructed by conducting a literature review from previously published papers. After the sustainability criteria are built through a literature review, then the list of criteria was clustered using the Kano model. This phase intended to develop the Kano cluster of sustainability criteria. There are five classifications in Kano: Must be, one dimensional, Attractive, Indifference, and Reverse. The chosen criteria for the next phase are those classified as "must be" categories. Fuzzy AHP then applied the "must be" criteria to determine the relative weights of each criterion. When the weight of each criterion is obtained, the next step is implementing the proposed method into a real case study in the palm oil industry to prove that it was valid and applicable. A decision matrix was used to select the most sustainable suppliers in this case study.

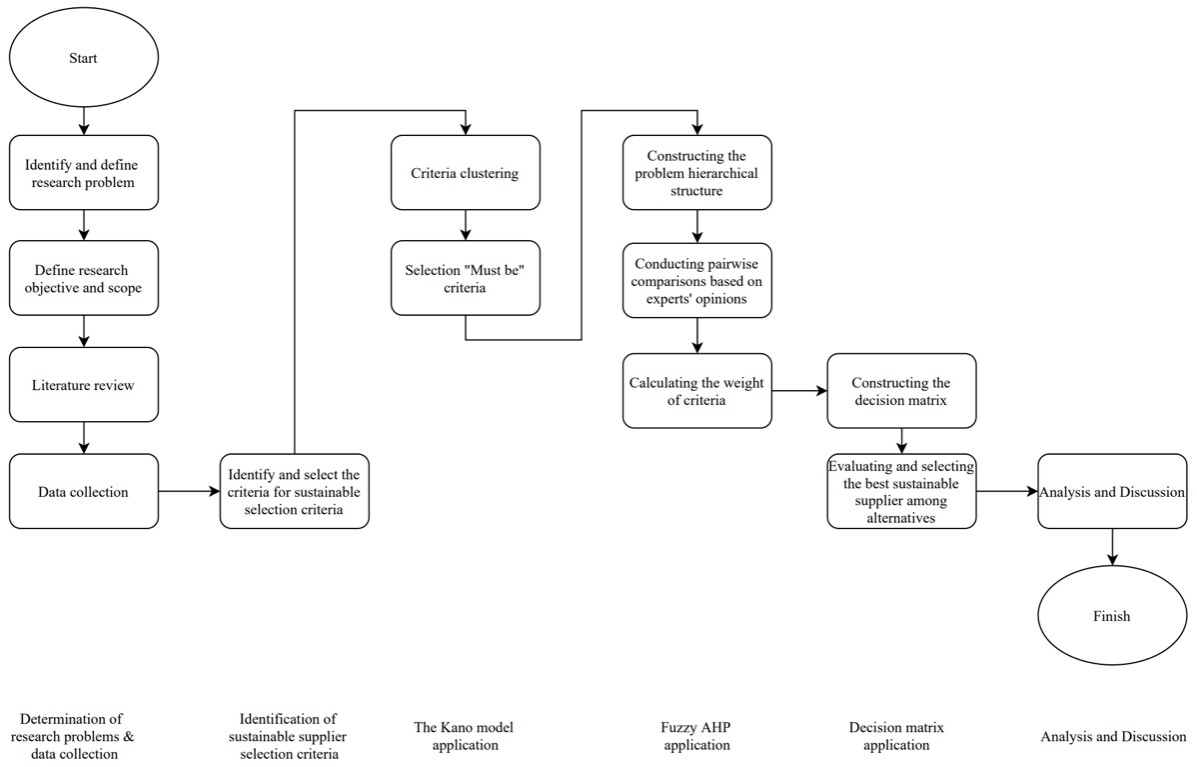

**Figure 1.** Research design.

### 3.3. Identification of Sustainable Supplier Selection Criteria

The criteria for sustainable supplier selection are constructed by conducting a literature review from previously published papers. Authors choose the criteria based on the most used criteria in supplier selection study, relevant to sustainable issues, and suitable to the palm oil industry. The selection of published papers as references is based on the citations. It relates to the impact factor of the journal and also the consideration that an enormous amount of citation describes the quality of a journal. Table 1 represents the sustainability criteria of this research, along with the description and references.

**Table 1.** Sustainability criteria for supplier selection.

| **Economic Sustainability Dimension** | | | |
|---|---|---|---|
| **No** | **Criteria** | **Description** | **References** |
| 1 | Quality | The ability of suppliers to provide products or services that meet customer expectations and satisfaction | [22,27,30,31,36,51–57] |
| 2 | Delivery | The ability of suppliers to deliver the products or materials needed on time and safely | [22,27,31,36,51,53–55,57] |
| 3 | Cost | The ability of a supplier to show all expenses related to the materials/goods supplied | [22,27,29–31,36,52,53,56–58] |
| 4 | Service | The supplier ensures the right service and able to solve the problems associated with supplied goods to the customer. | [22,27,31,36,52,53,55–57] |
| 5 | Responsiveness | The supplier is able to provide responses and feedback as soon as possible to the order. | [30,54] |
| 6 | Technical Capability | This factor shows the ability of supplier to solve the technical problem | [22,31,36,53,54,56] |
| 7 | Production facilities | Suppliers have appropriate production facilities and capacity requirements for a product. | [52,54,56,59] |
| 8 | Flexibility | Suppliers must be flexible enough to handle market variations. | [22,27,53,54,56,59] |
| 9 | Organization and Management | Suppliers should have an attitude towards improvements and cooperation, having strategic goals, risk-sharing capability, and openness in exchange information. | [54,56,60] |
| 10 | Financial | This factor shows the strength and stability of supplier's financial position, also supplier's growth. | [53,56,57] |
| 11 | Transportation cost | The tendency of suppliers to use minimal transportation cost for shipping the products. | [53] |
| 12 | The capacity of the supplier | Suppliers must have sufficient capacity to meet the company's needs. | [30,53,56] |
| 13 | Long term relationship | There is a long-term relationship between suppliers and companies. | [30,56] |
| 14 | Lead time | The supplier has a short time between the time when the order starts and being sent (waiting time). | [30,53] |
| 15 | Production technology | A supplier with new technology can respond to customers' orders quickly. | [30,53] |
| **Environmental Sustainability Criteria** | | | |
| **No** | **Criteria** | **Description** | **References** |
| 1 | Environmental management system | A set of organized processes and practices that allow a supplier to reduce its environmental impacts. | [27,30,31,36,53,54,56,57,61] |
| 2 | Energy consumption | This factor shows the effort of suppliers to use energy from renewable energy sources and do it efficiently. | [54,57,62] |
| 3 | Emissions | This factor shows suppliers considering the volume of carbon dioxide ($CO_2$) emissions and the volume of ozone-depleting emissions. | [29,54,57,62] |
| 4 | Water usage | Shows how suppliers consider the volume of wastewater produced and the degree of water pollution. | [54,57,62] |
| 5 | Waste | Shows the effort of a supplier to reduce the produced waste (solid, liquid, and hazardous waste). | [52,54,57,62] |

**Table 1.** *Cont.*

| 6 | Eco-design | Designing product with consideration of environmental impacts throughout the whole product lifecycle, including the stages of procurement, manufacture, use, and disposal. | [20,27,56] |
|---|---|---|---|
| 7 | Pollution control | The effort of a supplier in controlling pollution. | [22,27,36,53,58,61] |
| 8 | Green technology | The effort of suppliers in creating products with green technology. | [27,36] |
| 9 | Green manufacturing | The efforts of suppliers to minimize the consumption of raw materials and energy in producing products. | [53,63] |
| 10 | Green packing and labeling | The effort of suppliers to consider the environmental effect in packaging and labeling processes. | [29,53] |
| 11 | Environmental costs | The efforts of suppliers to use raw materials efficiently in the process production so that pollution and waste are minimal. | [53,56] |
| 12 | Environmental competencies | The ability of suppliers to use environmentally friendly substances, applying clean technology, processes, practices, and methods. | [53,56,61] |
| 13 | Green R&D and innovation | The efforts of suppliers to perform green research and development and do innovation in processes, practices, and methods. | [53,56,61] |
| 14 | Green product | The effort of suppliers in creating and producing green products. | [22,27,30,56,61] |
| 15 | Carbon tax | The carbon tax is the tax collected for $CO_2$ emissions. Its goal is to protect the environment from global warming by reducing $CO_2$ emissions. | [55] |
| 16 | Green transportation | This factor shows the effort of a supplier to reduce environmental pollution while conveying the needed order. | [27] |

| | | Social Sustainability Criteria | |
|---|---|---|---|
| **No** | **Criteria** | **Description** | **References** |
| 1 | Health and safety system | The efforts of suppliers to protect their personnel through a health and safety management system. Management provides a safe and secure workplace for their employees. | [22,27,29,31,36,53–57,61] |
| 2 | The interest and rights of employees | Suppliers have a concern for the rights of employees to achieve long-term sustainability effectiveness. | [27,31,36,53,56,58,61] |
| 3 | Child and forced labor | Concerns with engagement in child trafficking and forced labor. | [54,57] |
| 4 | Wages and working hours | Concerns with average working time, fair wage level, and overtime compensation. | [54,57] |
| 5 | Stakeholder involvement | The bundle of valued outcomes of the level that the building and environment make a connection between people, making or adding opportunities for positive social interaction. | [36,54,55,57] |
| 6 | The rights of stakeholders | Suppliers have a concern for the rights of stakeholders. | [36,53,56] |
| 7 | Employment practices | The efforts to meet the needs of current and future employees. | [22,36,55] |
| 8 | Information disclosure | The efforts to give or provide information to customers and stakeholders about the materials used, carbon emissions, and waste generated during production. | [31,53,56] |
| 9 | Supportive activities | Supplier efforts to provide activities that support employee performance, such as religious activities, recreation, or gathering events. | [27,30] |

### 3.4. The Kano Model Application

Criteria was clustered using the Kano model. This phase intended to develop the Kano cluster of sustainability criteria. There are five classifications in Kano: Must be, One dimensional, Attractive, Indifference, Reverse. The chosen criteria for the next phase are those classified as "must be" categories. The procedure steps for this phase have been adopted based on a previous study [34]. The steps are summarized as follows:

Step 1: Establish an online Kano questionnaire.
Step 2: Distribute the Kano questionnaire to several respondents.
Step 3: Evaluate the filled Kano questionnaire by aligned each answer pair with the Kano evaluation table.
Step 4: Calculate membership degree for each criterion.
Step 5: Classify the criteria based on the results of statistical analysis of questionnaires from respondents. The criteria were classified based on the highest value of membership degree.
Step 6: Select "must be" criteria for the next phase.

### 3.5. Fuzzy AHP Application

Fuzzy AHP was applied to the "must be" criteria to determine the relative weights of each criterion. Based on the literature review, there are many Fuzzy AHP procedures used in previous research. For this study, the procedure is followed Chang's Fuzzy Extent AHP (1996) and summarized as follows:

Step 1: Establish the hierarchy for sustainable supplier selection.
Step 2: Build a pair-wise comparison matrix based on the decision maker's preferences
Step 3: Build the fuzzy comparison matrix and convert the crisp number to Triangular Fuzzy Number (TFN), as shown in Table 2
Step 4: Determine the consistency index to evaluate the consistency of the pairwise matrix.
Step 5: Calculate the weights of criteria and sub criteria by defuzzification and normalization.

**Table 2.** Classification of sustainability criteria through the Kano Model.

| Linguistic Variable | Crisp Number | Fuzzy Number ($l$, $m$, $u$) |
|---|---|---|
| Equally Important | 1 | (1, 1, 3) |
| Weakly Important | 3 | (1, 3, 5) |
| Strongly Important | 5 | (3, 5, 7) |
| Very Strongly Important | 7 | (5, 7, 9) |
| Extremely Important | 9 | (7, 9, 11) |

### 3.6. Decision Making Application

After the weight of each criterion is obtained, the next step is implementing the proposed method into a real case study in the palm oil industry to prove that it was valid and applicable. A decision matrix was used to select the most sustainable suppliers in this case study. The procedure for this phase is summarized as follows:

Step 1: Construct the decision matrix The decision matrix consists of alternatives supplier, sustainability criteria and their weights, and total score. The alternatives suppliers were placed at the top of each column, sustainability criteria and weights were located in the left of the column, and the overall score is situated in the bottom of the matrix.
Step 2: Determine values for each concept Each alternative is given a value according to the nine-point Likert scale.
Step 3: Calculate the total value for each concept First, each alternatives-criteria combination is calculated by multiplying the weight of the criteria with the value and recorded in the second subcolumn. Afterward, sum the total values as a total score.

Step 4: Interpret the results The result with the highest cumulative score presented an indication of the most sustainable suppliers.

## 4. Results and Discussion

### 4.1. Clustering the Sustainability Criteria Using Kano Model

After the sustainability criteria are built through a literature review, then the list of criteria was clustered using the Kano model. This phase intended to develop the Kano cluster of sustainability criteria. There are five classifications in Kano: Must be, One dimensional, Attractive, Indifference, and Reverse. Since no respondents selected Reverse categories then these categories are not shown in Table 3. The chosen criteria for the next phase are those classified as "must be" categories.

**Table 3.** Classification of sustainability criteria through the Kano Model.

| Dimension | Kano Categories | | | |
| --- | --- | --- | --- | --- |
| | **Must Be** | **One Dimensional** | **Attractive** | **Indifferent** |
| **Economy** | • Quality<br>• Delivery<br>• Production Facilities<br>• The capacity of the supplier | • Service<br>• Responsiveness<br>• Flexibility<br>• Organization &Management<br>• Financial<br>• Transportation Cost<br>• Long term relationship<br>• Production Technologies | - | • Cost<br>• Technical capability<br>• Transportation cost<br>• Lead time |
| **Environmental** | • Environmental Management System<br>• Pollution control<br>• Environmental cost | • Carbon tax<br>• Eco-design<br>• Emissions<br>• Environmental competencies<br>• Waste<br>• Green Product<br>• Green transportation<br>• Water usage | • Green manufacturing<br>• Green product<br>• Green technology<br>• Green transportation<br>• Green R&D and innovation<br>• Green packing and labeling | • Energy consumption |
| **Social** | • Health & Safety System<br>• The interest and right of employees<br>• Child and forced labor<br>• Information disclosure | • Wages and working hours<br>• Stakeholder involvement<br>• The rights of stakeholder<br>• Employment practices<br>• Supportive activities | - | - |

Four criteria have been identified as must-be criteria in the economic dimension, namely quality, delivery, production facilities, and capacity of the supplier. In the environmental dimension, three out of sixteen criteria have been classified as must be criteria, i.e., environmental management system, pollution control, and environmental cost; this result is in line with the study from Wang et.al., [21], where pollution control and environmental management systems become the key criteria as sustainable supplier selection. In the social dimension, four criteria have been categorized as must-be criteria: health and safety systems, the interest and rights of employees, child, and forced labor, and information. The summary result of criteria clustering is presented in Table 3.

According to the result in the economic dimension, quality and delivery were categorized as must be criteria, and this result aligned with the findings of Zimmer et al. [57] that those criteria are in the top ten key criteria for sustainable supplier selection. This result also aligned with the study from Wang et.al. [21] where the criteria selection for economic dimension consists of delivery and quality. Then production capacity and capacity of the supplier are included in the requirements needed to obtain a sustainable supplier in the palm oil industry. In the environmental dimension, the palm oil industry expects a supplier

who has a management system that concerns the environment and uses raw materials efficiently in the production processes as an effort to control pollution and waste. This criteria selection for environmental dimension shows that pollution control and environmental management systems are selected as criteria, thus highlighting the pollution problem from the palm oil industry that actually has a harmful impact and becomes an environmental concern [12].

In the social dimension, the palm oil industry expects a supplier who concerns the rights of their employees, does not employ underage force labor, and has a management system that provides safe workplaces, and manages the health of their employees. Health & safety is included as the selected criteria and this result is in line with the previous studies from Yang and Wang [64] and also Wang et.al., [21]. Moreover, suppliers should provide important information during the process of production, such as materials used, carbon emissions and waste generated, etc. All these criteria are assumed to be able to achieve long-term sustainability.

### 4.2. Generating "Must Be" Criteria Weights Using Fuzzy AHP

After the criteria were clustered through Kano and 'Must be' criteria were chosen, the next phase is to determine their weights by fuzzy AHP. First of all, the hierarchy for this study was structured. Figure 2 illustrates the hierarchy of the sustainable supplier selection problem in four levels.

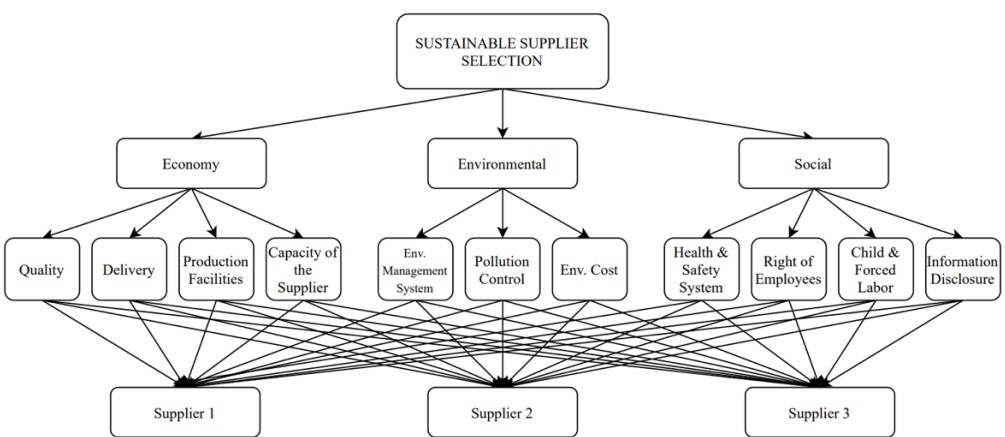

**Figure 2.** Hierarchy of the sustainable supplier selection.

In this research, Chang's fuzzy extent AHP was used to determine the weights of the criteria. The author constructs an aggregate pairwise comparison matrix and distributes it to several experts to obtain the weights of each criterion. The results are summarized in Table 4.

**Table 4.** The weight of the criteria and sub-criteria examined by experts.

| Criteria | Local Weight | Sub-Criteria | Local Weight | Global Weight |
|---|---|---|---|---|
| **Economy** | 0.3862 | Quality | 0.3465 | 0.1338 |
| | | Delivery | 0.3056 | 0.1180 |
| | | Production Facilities | 0.0946 | 0.0365 |
| | | The capacity of the supplier | 0.2533 | 0.0978 |
| **Environmental** | 0.4332 | Env. Management System | 0.3431 | 0.1486 |
| | | Pollution Control | 0.3793 | 0.1643 |
| | | Environmental Cost | 0.2776 | 0.1203 |
| **Social** | 0.1805 | Health & Safety System | 0.2444 | 0.0441 |
| | | The Interest & Right of Employee | 0.2444 | 0.0441 |
| | | Child & Forced Labor | 0.2444 | 0.0441 |
| | | Information Disclosure | 0.2668 | 0.0482 |

Table 4 represents the local and global weight for all criteria in this research. In the three pillars of sustainability, the environmental dimension has the first rating with a local weight of 0.4332, followed by the economy and social dimension with local weight 0.3862 and 0.1805, respectively. It shows that for the palm oil industry, the environmental dimension is the most priority in selecting a sustainable supplier. Hence, this result is contrary to the study from Yang and Wang [64], where the economic dimension becomes the highest weight followed by the environmental and social dimension. This probably comes from the expert judgement weighting result, because it depends on the pairwise comparison. Therefore, there's also the possibility of weight and rating differences when the set of criteria changes.

In economic criteria, quality is the highest rating with a local weight 0.3465, whereas production facilities ranked lowest with a local weight 0.0946. Pollution control has a local weight 0.3793 and becomes the priority in the environmental criteria. Furthermore, information disclosure has the highest importance in social criteria. While the remaining criteria, health and safety systems, the interest and rights of an employee, and child and forced labor have the same local weight that is 0.2444. The detailed results can be seen in Appendix B.

A consistency test is carried out to evaluate that the expert's opinion is consistent or inconsistent. Each pairwise comparison matrix was tested, and the consistency ratio must be less than 0.10. If the consistency ratio value is more than 0.10, that means the result is inconsistent. Thus, experts must fill out the questionnaire again until consistent.

Table 5 shows the result of the consistency test for each pairwise comparison matrix. It can be seen that the consistency ratio for all hierarchies is less than 0.10. The consistency ratio for sustainability dimension, economy, and social hierarchy is 0.0034; 0.0048; 0.0246, respectively. The environmental hierarchy has a very consistent result with a consistency ratio: 0.0000. According to that, it can be concluded that each pairwise comparison matrix in this research is consistent.

**Table 5.** Consistency computation for each pairwise comparison matrix.

| Hierarchy | Items | Values |
|---|---|---|
| **Sustainability dimension** | Maximum Eigenvalue (λmax) | 3.0039 |
| | Consistency Index (CI) | 0.0019 |
| | Random Index (RI) n = 3 | 0.5800 |
| | Consistency Ratio (CR) | 0.0034 |
| **Economy criteria** | Maximum Eigenvalue (λmax) | 4.0129 |
| | Consistency Index (CI) | 0.0043 |
| | Random Index (RI) n = 4 | 0.9000 |
| | Consistency Ratio (CR) | 0.0048 |
| **Environmental criteria** | Maximum Eigenvalue (λmax) | 3.0000 |
| | Consistency Index (CI) | 0.0000 |
| | Random Index (RI) n = 3 | 0.5800 |
| | Consistency Ratio (CR) | 0.0000 |
| **Social criteria** | Maximum Eigenvalue (λmax) | 4.0665 |
| | Consistency Index (CI) | 0.0222 |
| | Random Index (RI) n = 4 | 0.9000 |
| | Consistency Ratio (CR) | 0.0246 |

*4.3. Implementation in Real World Using Decision Matrix*

After the framework of a sustainable supplier selection has been constructed, the author implements the proposed framework to show the applicability and usefulness to the real world. In this study, one of the palm oil industries in Indonesia was chosen to apply the proposed framework. Due to confidentiality, the name of the company is prohibited from being revealed and called PT. XYZ. This company is a palm oil producer that has

seven system management, which are ISO 9001:2008, ISO 14001:2004, OHSAS 18001:2007, ISO 22000:2005, ISPO, Good Corporate Governance, and Malcolm Baldridge.

PT. XYZ then evaluated and selected the most sustainable supplier from three alternatives suppliers using the proposed framework in the form of a decision matrix. In this case, the suppliers were referred to supplier 1, supplier 2, and supplier 3. All the suppliers supplied spare parts for the company. Three decision-makers from the procurement department were asked to assess and evaluate all the suppliers. All of them have interaction with all the suppliers and have a responsibility for the company's procurement decisions.

Table 6 shows the result of the evaluated three suppliers by decision-makers using a decision matrix. The value for each supplier is the average value based on the evaluation of all decision-makers (Appendix C). From the table, it can be seen that the highest total value was on supplier 1 with the amount of 23.6503. Following that, supplier 3 has a total value of 22.9884, while supplier 2 has the lowest overall value, which was only 21.8961.

**Table 6.** Decision Matrix Result.

| No. | Sustainability Dimension | Criteria | Weight | Value Supplier 1 | Supplier 2 | Supplier 3 |
|---|---|---|---|---|---|---|
| 1 | Economy | Quality | 0.3465 | 8.0000 | 7.6667 | 8.3333 |
| | | Delivery | 0.3056 | 8.0000 | 7.3333 | 8.3333 |
| | | Production facilities | 0.0946 | 8.6667 | 6.6667 | 7.3333 |
| | | The capacity of the Supplier | 0.2533 | 8.6667 | 7.6667 | 7.6667 |
| 2 | Environmental | Env. Management system | 0.3431 | 7.3333 | 7.0000 | 7.3333 |
| | | Pollution control | 0.3793 | 7.3333 | 7.0000 | 7.3333 |
| | | Env. Cost | 0.2776 | 7.6667 | 7.3333 | 7.3333 |
| 3 | Social | Health and Safety System | 0.2444 | 8.3333 | 7.3333 | 8.0000 |
| | | The Interest and right of the employees | 0.2444 | 8.0000 | 7.3333 | 7.3333 |
| | | Child and force labor | 0.2444 | 8.0000 | 7.3333 | 7.3333 |
| | | Information disclosure | 0.2668 | 7.6667 | 7.3333 | 7.6667 |
| | | **Total Score** | | **23.6503** | **21.8961** | **22.9884** |

Based on the total value in Table 5, supplier 1 is the most sustainable compared to the other suppliers. Even based on Figure 3, it shows that supplier 1 has excellent performance for all sustainability dimensions. It means that choosing and selecting supplier 1 to be the partner and work-with will increase the company's sustainability. On the other hand, supplier 3 was on the second rank for sustainable performance. The value for economic, environmental, and social are 8.0699, 7.3333, and 7.5852, respectively.

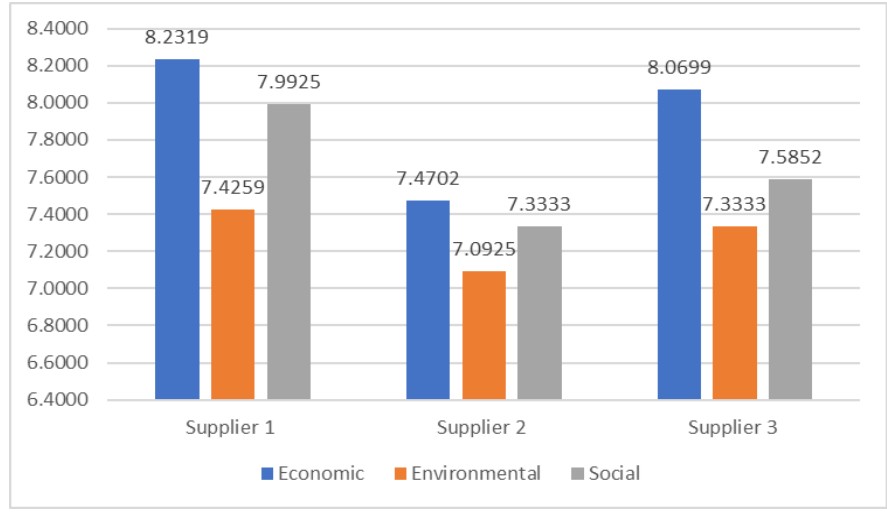

**Figure 3.** Supplier's value for each sustainability dimension.

Furthermore, supplier 2 has the lowest performance for all sustainability dimensions. It can be seen from their value for each sustainability dimension. This shows that, supplier 2 needs to be concerned about their performance in economic criteria, increase their effort to protect the environment, and have to take action to improve their social dimension performance.

After the result is obtained, the author confirms the result and checks the agreement to all respondents by showing the results of the decision matrix. The three respondents agreed with the results that supplier 1 is the most sustainable compared to the other suppliers. They stated that the result is aligning with the experience and actual condition while they had interaction with the supplier. Besides that, the respondents also said the form of the decision matrix is easy to fill and applicable to be used in their company.

The framework can be used for making a decision about the most sustainable supplier. In addition, the proposed model provides a view of the company's decision-maker for decision making and can be a reference for a supplier to improve its performance. The form of a decision matrix also offers easiness and simplicity in application.

## 5. Conclusions

This research aimed to develop a framework for selecting a sustainable supplier in the palm oil industry. To establish the framework, this study proposed a novel integrated method using the Kano model, Fuzzy Analytical Hierarchy Process, and Decision Matrix. The initial process started by identifying the sustainability criteria by conducting a literature review. From that process, forty sustainability criteria were obtained, consisting of fifteen criteria for economic, nine criteria for social, and sixteen criteria for environmental.

The Kano model was applied for criteria clustering. It was found that quality, delivery, production facilities, and capacity of the supplier were clustered as "must be" criteria for the economic dimension. In the environmental dimension, a management system, pollution control, and environmental cost as "must be" criteria were obtained. Further, in the social dimension, there are health and safety systems, the interest and rights of employees, child and forced labor, and information disclosure categorized as "must be" criteria. Fuzzy AHP was used to determine the weight of each criterion, and the Decision Matrix Method was performed for selecting the most sustainable supplier. As Indonesia's government established ISPO policy concerning environmental issues, the Kano model provides results that support ISPO. The selected criteria in the "must have" category in the process of selecting a sustainable supplier are environmental management systems and pollution. These criteria are often mentioned in several studies in the environmental dimensions. Not only that, in the social dimensions, health and safety were also become the selected criteria. This criterion is one of the main keys of this dimension.

The Fuzzy AHP result shows that the environmental dimension is the most sustainable supplier criteria, followed by economy and social dimension. Quality, pollution control, and information disclosure were found to be important sub-dimension for sustainable supplier selection. These criteria could help the palm oil industries to achieve sustainable supplier selection. A limitation of this study is that there will be differences in weights and ratings when the set of criteria or expert changes. However, the main contribution is to generate a novel method for selecting a sustainable supplier. For future research, it is possible to build a methodology by integrating the qualitative model, mathematical analytical, and artificial intelligence. Other than that, this research could be applied to different industries for a case study.

**Author Contributions:** Conceptualization, T.-C.K. and K.H.F.; methodology, T.-C.K. and K.H.F.; writing—original draft preparation, K.H.F. and M.M.; writing—review and editing, T.-C.K. and M.M.; All authors have read and agreed to the published version of the manuscript.

**Funding:** This research received no external funding.

**Institutional Review Board Statement:** Not applicable.

**Informed Consent Statement:** Not applicable.

**Data Availability Statement:** The data presented in this study are available on request from the corresponding author.

**Conflicts of Interest:** The authors declare no conflict of interest.

## Appendix A

**Table A1.** The Kano Evaluation Table.

| No | Economy Dimension Criteria | Respondent | | | | | | | | | | | | | | | | | | | | | | | | | | | | | | | | |
|----|------|---|---|---|---|---|---|---|---|---|---|---|---|---|---|---|---|---|---|---|---|---|---|---|---|---|---|---|---|---|---|---|---|---|
| | | 1 | 2 | 3 | 4 | 5 | 6 | 7 | 8 | 9 | 10 | 11 | 12 | 13 | 14 | 15 | 16 | 17 | 18 | 19 | 20 | 21 | 22 | 23 | 24 | 25 | 26 | 27 | 28 | 29 | 30 | 31 | 32 | 33 |
| 1 | Quality | M | O | M | A | O | M | M | I | M | I | I | M | O | O | M | O | I | O | M | M | I | M | M | M | M | O | O | O | O | M | M | M | O |
| 2 | Delivery | M | O | M | A | O | M | M | I | O | I | O | M | O | O | M | M | I | O | M | M | I | O | M | M | M | O | O | O | O | M | I | I | I |
| 3 | Cost | M | O | I | I | O | M | O | O | M | I | O | I | I | I | A | M | I | I | I | M | M | O | I | M | M | O | O | M | R | M | I | R | O |
| 4 | Service | Q | O | I | A | O | I | O | A | O | M | A | M | O | O | O | M | I | O | M | M | M | O | M | M | M | O | O | O | A | O | I | M | O |
| 5 | Responsiveness | I | O | I | A | O | O | O | I | O | M | O | M | O | M | O | O | M | M | I | O | M | M | M | M | O | M | M | O | O | O | O | M | I | M | M |
| 6 | Flexibility | M | A | I | A | O | O | O | A | O | M | O | I | I | I | A | M | I | A | I | I | O | M | O | M | M | O | O | O | I | I | I | A | I |
| 7 | Production facilities | M | O | I | A | O | M | I | I | M | M | I | M | O | M | I | M | I | A | M | M | I | M | M | I | M | O | O | O | M | I | I | O | O |
| 8 | Technical capability | M | O | I | A | O | M | I | I | M | M | I | A | O | O | I | M | I | O | M | I | I | A | M | I | M | O | O | O | I | O | I | I | A |
| 9 | Organization and management | M | A | I | A | O | O | M | O | I | O | O | A | O | I | I | M | I | O | M | I | M | I | O | M | M | O | O | O | M | M | A | A | A |
| 10 | Financial | M | O | I | A | O | M | O | A | M | O | O | A | I | I | I | O | M | O | M | I | M | O | M | M | M | O | O | O | I | O | O | A | I |
| 11 | Transportation cost | M | O | I | A | O | O | I | M | M | M | I | A | I | A | A | M | I | A | I | M | I | I | A | M | M | O | O | O | A | A | I | I | O |
| 12 | Capacity of the supplier | M | O | I | A | O | M | I | I | M | I | O | A | O | I | I | M | I | O | M | M | I | O | M | M | M | O | O | O | I | M | O | M | I |
| 13 | Long term relationship | I | O | I | I | O | O | O | I | O | I | O | A | M | A | O | M | I | A | I | M | I | O | I | I | M | O | O | O | A | M | O | A | M |
| 14 | Lead time | A | O | I | A | O | M | A | A | M | I | I | A | I | A | M | M | I | O | I | I | M | M | O | I | M | O | O | A | I | I | O | M | I |
| 15 | Production technologies | M | O | I | A | O | O | A | O | A | I | O | I | O | A | I | M | I | O | M | M | I | I | I | I | M | O | O | O | I | O | I | O | A |

Calculate the maximum kano categories (M: Must be; O: One dimensional; A: Attractive; and I: Indifferent) then selected only "Must be" categories to become criteria of sustainable supplier selection.

## Appendix B

**Table A2.** Integrated Fuzzy Comparison Matrix Each Dimension.

| | Economy | | | Environmental | | | Social | | |
|---|---|---|---|---|---|---|---|---|---|
| **Economy** | 1 | 1 | 1 | 0.6776 | 0.7248 | 1.9332 | 1.5518 | 2.3714 | 4.6632 |
| **Environmental** | 0.5173 | 1.3797 | 1.4758 | 1 | 1 | 1 | 1.9037 | 2.7131 | 5.1563 |
| **Social** | 0.2144 | 0.4217 | 0.6444 | 0.1939 | 0.3686 | 0.5253 | 1 | 1 | 1 |

**Table A3.** Fuzzy Extent AHP.

| Dimension | Fuzzy Sum of Each Row | | | Fuzzy Synthetic Extent | | | Degree of Possibility of Si ≥ Sj | | | Degree of Possibility (Si) | Normalization |
|---|---|---|---|---|---|---|---|---|---|---|---|
| **Economy** | 3.2295 | 4.0962 | 7.5964 | 0.1856 | 0.3731 | 0.9426 | | 0.8915 | 1 | 0.8915 | 0.3863 |
| **Environmental** | 3.4209 | 5.0928 | 7.6321 | 0.1966 | 0.4639 | 0.9471 | 1 | | 1 | 1 | 0.4332 |
| **Social** | 1.4084 | 1.7903 | 2.1697 | 0.0810 | 0.1631 | 0.2692 | 0.6000 | 0.4166 | | 0.4166 | 0.1805 |

# Appendix C

**Table A4.** Calculation of Decision Matrix.

| No | Sustainability Dimension | Criteria | Weight | Supplier 1 | | | Average | W x Value | Supplier 2 | | | Average | W x Value | Supplier 3 | | | Average | W x Value |
|----|--------------------------|----------|--------|----|----|----|---------|-----------|----|----|----|---------|-----------|----|----|----|---------|-----------|
| | | | | R1 | R2 | R3 | | | R1 | R2 | R3 | | | R1 | R2 | R3 | | |
| 1 | Economy | Quality | 0.35 | 8 | 8 | 8 | 8.00 | 2.77 | 8 | 7 | 8 | 7.67 | 2.66 | 9 | 7 | 9 | 8.33 | 2.89 |
| | | Delivery | 0.31 | 8 | 8 | 8 | 8.00 | 2.44 | 7 | 6 | 9 | 7.33 | 2.24 | 9 | 7 | 9 | 8.33 | 2.55 |
| | | Production Facilities | 0.09 | 9 | 9 | 8 | 8.67 | 0.82 | 7 | 6 | 7 | 6.67 | 0.63 | 7 | 7 | 8 | 7.33 | 0.69 |
| | | The capacity of the Supplier | 0.25 | 9 | 9 | 8 | 8.67 | 2.20 | 8 | 7 | 8 | 7.67 | 1.94 | 8 | 7 | 8 | 7.67 | 1.94 |
| 2 | Environmental | Env. Management System | 0.34 | 8 | 7 | 7 | 7.33 | 2.52 | 8 | 6 | 7 | 7.00 | 2.40 | 8 | 6 | 8 | 7.33 | 2.52 |
| | | Pollution Control | 0.38 | 8 | 7 | 7 | 7.33 | 2.78 | 8 | 6 | 7 | 7.00 | 2.66 | 8 | 6 | 8 | 7.33 | 2.78 |
| | | Environmental Cost | 0.28 | 8 | 7 | 8 | 7.67 | 2.13 | 8 | 6 | 8 | 7.33 | 2.04 | 8 | 6 | 8 | 7.33 | 2.04 |
| 3 | Social | Health and Safety System | 0.24 | 8 | 8 | 9 | 8.33 | 2.04 | 8 | 6 | 8 | 7.33 | 1.79 | 8 | 7 | 9 | 8.00 | 1.96 |
| | | The Interest & Right of Employees | 0.24 | 8 | 8 | 8 | 8.00 | 1.96 | 8 | 6 | 8 | 7.33 | 1.79 | 8 | 6 | 8 | 7.33 | 1.79 |
| | | Child and Forced Labor | 0.24 | 8 | 8 | 8 | 8.00 | 1.96 | 8 | 6 | 8 | 7.33 | 1.79 | 8 | 6 | 8 | 7.33 | 1.79 |
| | | Information Disclosure | 0.27 | 8 | 7 | 8 | 7.67 | 2.05 | 8 | 6 | 8 | 7.33 | 1.96 | 8 | 6 | 9 | 7.67 | 2.05 |
| | TOTAL VALUE | | | | | | | 23.65 | | | | | 21.90 | | | | | 22.99 |

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
