# Peer review of "An Integrated Kano Model, Fuzzy Analytical Hierarchy Process, and Decision Matrix for Sustainable Supplier Selection in Palm Oil Industries Indonesia, a Case Study"

_processes, doi:10.3390/pr9061078_

Round 1

Reviewer 1 Report

  1. This manuscript is about developing a framework for selecting a sustainable supplier in the palm oil industry. The topic is relevant to the journal's aims. It is informative and interesting for the target audience, however, it has serious faults as the research process and results were not clearly presented.
  2. Poor research design, scanty description of methods and process: It is not clear how and why the authors chose the used methods. The authors explain why they use MCDM method clearly in the article, however, it is not clear how they chose to use Kano model, Analytic Fuzzy Hierarchy Process (AHP). How did they conclude that these are the best methods? Which other methods did they explore and test before choosing these methods?
  3. The process needs to be explained elaborately. Now, this information is hardly available. This also means that a lot of information is lacking that is needed if somebody wants to replicate this study.
  4. The discussion part of the manuscript does not comply with the aims and rules of a discussion part of a scientific paper. The authors should compare their framework with the other existing methods and frameworks for example the ones that they mention in Section 2, paragraph 4.
  5. The English of this manuscript needs to be checked by a professional as it includes many mistakes. 
  6. The authors can mention the environmental problems related to palm oil production briefly with 3-4 sentences.
  7. Section 1, 4th paragraph needs more references. The authors talk about the policy that forces palm oil industries in Indonesia to implement sustainability into their business. 
  8. Section 2, paragraph 4: A more detailed description of Kano model is needed before writing about its advantages and disadvantages for the companies.
  9. Repetitions make it hard to read the manuscript. For example, the sentences and arguments in paragraph 8 in Section 4 are repeated.
  10. Some sentences in the manuscript do not seem necessary and meaningful for example, Section 3.1 first sentence. Section 1 last paragraph is not necessary.

Reviewer 2 Report

The conducted research is very interesting and multi-threaded. Below please find a few comments helpful in improving the manuscript:
1.    Common practice in scientific papers is to include a brief paragraph at the end of the Introduction or Research methods in order to indicate the structure of the document. This helps the reader to have an accurate idea about the organization and facilitates the reading.  In order to understand the entire context of the research, please present the stages of the conducted research or a respective flowchart.
2.    The document is very concise and facilitates the reader's comprehension but it must contain some graphic image that helps to the comprehension of the tables (figure).
3. Has the author used any feature scoring (grading)?
(concerns: criteria for sustainable supplier selection)
3.    Discussion of other authors is good but withaut references in part 5. Discussion.
4. Please distinguish and point out the conclusions

Reviewer 3 Report

Article in the first part written correctly, abstract correct, reflecting the content of the publication. Objective and correctly conducted literature analysis. According to the law of large numbers, 33 questionnaires is the minimum number. The reviewer suggests increasing the sample size, which would make the research more credible. As we read in the publication, the authors are aware of these relationships. Sufficient presentation of results and their description. The selection of 15 economic, 9 social and 16 environmental characteristics respectively is not explained. Surprising is the shallow discussion of the results, which is limited to a re-description of the methodology without reference to the results and findings of other studies. With changes, the paper is suitable for publication in a journal.
